structural engineering/mechanical engineering/mechanics

bistability, polar orthotropy, nonlinear shell theory, morphing structures, post-buckling analysis, analytical approach

**Author for correspondence:**
K. A. Seffen
e-mail: kas14@cam.ac.uk

# Bistable polar-orthotropic shallow shells

## P. M. Sobota and K. A. Seffen

Department of Engineering, University of Cambridge, Trumpington Street, Cambridge CB2 1PZ, UK

PMS, 0000-0002-6294-8329; KAS, 0000-0002-7725-0361

We investigate stabilizing and eschewing factors on bistability in polar-orthotropic shells in order to enhance morphing structures. The material law causes stress singularities when the circumferential stiffness is smaller than the radial stiffness ($\beta < 1$), requiring a careful choice of the trial functions in our Ritz approach, which employs a higher-order geometrically nonlinear analytical model. Bistability is found to strongly depend on the orthotropic ratio, $\beta$, and the in-plane support conditions. An investigation of their interaction offers a new perspective on the effect of the hoop stiffness on bistability: while usually perceived as promoting, it is shown to be only stabilizing insofar as it prevents radial expansions; however, if in-plane supports are present, it becomes a redundant feature. Closed-form approximations of the bistable threshold are then provided by single-curvature-term approaches. For significantly stiffer values of the radial stiffness, a strong coupling of the orthotropic ratio and the support conditions is revealed: while roller-supported shells are monostable, fixed-pinned ones are most disposed to stable inversions; insight is given by comparing to a simplified beam model. Eventually, we show that cutting a central hole is a suitable method to deal with stress singularities: while fixed-pinned shells are barely affected by a hole, the presence of a hole strongly favours bistable inversions in roller-supported shells.

## 1. Introduction

Shells with more than one stable equilibrium state have enjoyed considerable interest in the engineering community, since their *multistability* enables them to adapt to changing loading conditions in beneficial ways. Examples in engineering applications range from aerofoils [1,2], flow regulators [3] and deployable structures [4–6] to novel semiconductor production processes [7], motors [8], nanofilms [9,10], wind-turbine blades [11], piezoelectric energy harvesting devices [12,13] and several other microelectromechanical systems [14,15].

**Figure 1.** Initial stress free shells (top), their stable inversions (middle) and a sketch of their profiles (bottom). (*a*) Isotropic shell with an approximately uniformly curved counterpart. (*b*) Circumferentially stiffened shell ($\beta > 1$), mimicking globally an orthotropic shell, since radial stresses cannot efficiently build up in the stiffened regions. The inverted configuration exhibits a central dimple, which becomes even more distinct, when the orthotropic ratio is increased, cf. (*c*). (*d*) Radially stiffened shell ($\beta < 1$) with a central plug due to the manufacturing process; its inversion evinces concentrated deformations at the centre.

Bistability in shells is engendered by them having initial, usually positive Gaussian curvature [7,16–19], being made from non-isotropic materials [17,18,20–23], or by pre-stressing [24–26]; see [27] for basic design criteria.

Interestingly, most structures possess a uniform thickness profile, even though local thickness variations are commonly employed by engineers to improve the structural behaviour. Structures with such variations include corrugated and dimpled shells that globally mimic anisotropic constitutive laws due to a smeared stiffness [28,29]. Other local phenomena such as creasing and grooving [30] were recently shown to cause bistability despite not precisely belonging to one of the methods above.

A simple series of table top experiments with cast silicon rubber shells in figure 1 illustrates that radial and circumferential stiffeners strongly influence the way in which initially uniform caps invert; it is notable that also the minimum height required for a bistable inversion differs significantly. Hence, it is possible to gain control over bistable behaviour by making simple cross-sectional adjustments, e.g. by using appropriate stiffeners, grooves or employing similar structures, such as grid-shells, which ultimately offer the opportunity to exploit this characteristic in an optimized manner.

We formally analyse the influence of such manipulations on bistability by considering a polar-orthotropic constitutive law—an aspect which to our knowledge is completely novel. An analytical approach is undertaken, since it provides further insight into the governing factors. Coulais [31] recently identified the lack of such models as a bottleneck in the development of novel smart structures, since alternative methods, such as the finite-element (FE) method, cannot identify geometrical thresholds without undertaking tedious numerical parameter studies; this study can be regarded as a direct response to this need.

Polar orthotropy allows us to vary the *internal* directional stiffness of shells, which sheds light into the statically indeterminate interplay between radial and hoop stiffness, and points towards optimized values that stabilize (or diminish) bistable inversions. This knowledge enables us to make bistable shells more efficient, to save material, and to make such structures more versatile by allowing the tailoring of the shape of alternative equilibrium configurations; furthermore, it enables us to judge where the unique features of shells are required and where simpler beam structures suffice.

The interaction of the stiffness ratio, $\beta$, with the in-plane support conditions is of particular interest in this study, since extreme ratios of $\beta$ may lead to a predominantly uniaxial load path; the bistable response of one-dimensional structures, however, strongly depends on additional horizontal supports. This additionally widens the applicability of bistable structures by considering support conditions that are commonly encountered in practice.

We consider shells with a circular planform that are free to rotate around the outer rim support with a variable radial spring stiffness, $K_u$, enabling roller-supported edges ($K_u = 0$) to fixed-pinned supports ($K_u \to \infty$). We neglect the details of the transition between equilibria, which may include non-axisymmetric secondary buckling modes [32], and aim to find the inverted axisymmetrical configurations accurately.

Convoluted expressions resulting from applying Bessel functions [33] are avoided by employing polynomials in a variational higher-order approach, with up to three degrees of freedom for caps and annuli. Polynomials expedite an analytical treatment and, even though they do not precisely satisfy the equilibrium conditions [34], they give sufficient accuracy.

The trial functions employed—which are of real but, in general, not integer order—incorporate a novel extension for higher-order approaches that is based on the geometrically linear solution of a bent plate, in order to address the stress singularities arising in shells with an increased radial stiffness ($\beta < 1$). While this parameter-range is often neglected or otherwise circumvented, e.g. by introducing a central hole or an isotropic plug [35–37], we address this problem directly and use it to demonstrate the robustness of our approach. In particular, we compare the stress resultants to FE results and highlight that shape functions of integer order occasionally employed in other approaches in the literature inevitably fail to capture stress singularities in bending and, hence, dangerously *underestimate* stress-levels.

We then analyse the effects of the stiffness ratio, $\beta$, as well as the influence of additional horizontal supports on the required apex height for bistable inversion. We also present a drastic simplification of our model by using a single degree of freedom, which enables us to approximate the threshold of bistability in closed form. The results are supported by considering simpler beam structures that elucidate governing factors of bistability for extreme ratios of $\beta$.

Finally, we introduce a central hole and investigate its influence on bistability, in order to allow for real applications of radially stiffened shells where stress singularities are usually not acceptable. In the following section, the governing equations are given before the analytical model is derived in §3. The results are presented in §4 and, eventually, a summary and outlook are given in §5.

**Remark 1.1** Polar orthotropy differs from the often-studied rectilinear orthotropy [18,20,22] since the possible misalignment of principal strain directions and material-orientations in the latter evokes a strain-energy performance more conducive to forming extra stable equilibrium configurations. In polar-orthotropic materials, however, this misalignment is not observed as long as rotational symmetry is preserved because the absence of in-plane shear is tantamount to principal strains that align with the principal material-orientations. Hence, we do not expect to observe additional stable configurations stemming from the material law.

## 2. Governing equations

The $z$-ordinate of a rotationally symmetric, initially curved but stress-free shallow shell, with an outer planform radius $a$ and an initial midpoint deflection $w_0^M$ is described in cylindrical coordinates ($r$, $\theta$, $z$) by a function $w_0 = (1 - r^2/a^2)w_0^M$. The nonlinear response of this shell can be described by two potential functions, $w$ and $\Phi$. The first describes the deflection and is the potential of the change in radial, circumferential and twisting curvature

$$\kappa_r = -w'', \quad \kappa_\theta = -\frac{w'}{r} \quad \text{and} \quad \kappa_{r\theta} = 0, \tag{2.1}$$

respectively, where a prime denotes a derivative with respect to $r$. The second potential is the Airy stress function, $\Phi$, and relates to in-plane stresses via:

$$\sigma_r = \frac{\Phi'}{r}, \quad \sigma_\theta = \Phi'' \quad \text{and} \quad \sigma_{r\theta} = 0. \tag{2.2}$$

The relationship to strains is established via a polar-orthotropic constitutive law from three independent parameters: $E$, $v$ and $\beta$; $E = E_r$ denotes Young's modulus in the radial direction; $v = v_{\theta r}$, Poisson's ratio, and $\beta$ is the orthotropic ratio. Note that Poisson's ratio is not symmetric with respect to indices, since an associated lateral contraction depends on the stiffness ratio, and their compliance requires $E_r v_{\theta r} = E_\theta v_{r\theta}$. Hence, the stiffness ratio

$$\beta = \frac{E_\theta}{E_r} \tag{2.3}$$

also describes the ratio of Poisson's effects in particular directions via $\beta = v_{\theta r}/v_{r\theta}$; see [33,38] for details.

Due to rotational symmetry, the constitutive equations are written in terms of the principal directions only. The corresponding material tensor, $\mathbf{E}$, is thus a $2 \times 2$ matrix:

$$\mathbf{E} = \frac{E}{1 - v^2/\beta} \begin{bmatrix} 1 & v \\ v & \beta \end{bmatrix} \quad \text{and} \quad \mathbf{E}^{-1} = \frac{1}{E} \begin{bmatrix} 1 & -v/\beta \\ -v/\beta & 1/\beta \end{bmatrix}, \tag{2.4}$$

where positive definiteness sets $\beta > v^2$. Integrating in the thickness direction gives the stretching and flexural rigidity, and for ease of notation we introduce $D = Et^3\beta/12(\beta - v^2)$. Now, the corresponding bending stresses and in-plane strains read as

$$\begin{bmatrix} m_r \\ m_\theta \end{bmatrix} = D \begin{bmatrix} 1 & v \\ v & \beta \end{bmatrix} \begin{bmatrix} \kappa_r \\ \kappa_\theta \end{bmatrix} \quad \text{and} \quad \begin{bmatrix} \varepsilon_r \\ \varepsilon_\theta \end{bmatrix} = \frac{1}{E} \begin{bmatrix} 1 & -v/\beta \\ -v/\beta & 1/\beta \end{bmatrix} \begin{bmatrix} \sigma_r \\ \sigma_\theta \end{bmatrix}, \tag{2.5}$$

respectively.

Equilibrium considerations require the balance of moments and vertical forces, which state:

$$\frac{m_\theta - m_r}{r} - m_r' = q_r \quad \text{and} \quad \left(\frac{\mathrm{d}}{\mathrm{d}r} + \frac{1}{r}\right) q_r = -p - \frac{1}{r}(\Phi'(w' + w_0'))', \tag{2.6}$$

where $q_r$ denotes the shear force and $p$ a transverse pressure loading. By combining equations (2.1), (2.5) and (2.6), we express these equilibrium conditions in terms of $w$ and $\Phi$:

$$w'''' + \frac{2}{r} w''' - \frac{\beta}{r^2} w'' + \frac{\beta}{r^3} w' = \frac{1}{D}\left[p + \frac{t}{r}(\Phi'(w' + w_0'))'\right]. \tag{2.7}$$

From the Airy stress function, we ensure that the in-plane equilibrium condition, $\sigma_r - \sigma_\theta + r\sigma_r' = 0$, is satisfied for arbitrary choices of $\Phi(r)$. Besides equilibrium, the geometrically nonlinear strain definitions of

$$\varepsilon_r = u' + \frac{1}{2}(w' + w_0')^2 - \frac{1}{2}(w_0')^2 \quad \text{and} \quad \varepsilon_\theta = \frac{u}{r} \tag{2.8}$$

require compatibility, which is enforced by eliminating the radial displacement, $u$, in the preceding equation by using $u' = (r\varepsilon_\theta)'$. After substituting $\varepsilon_r$ and $\varepsilon_\theta$ according to equation (2.5) and again with equation (2.2), the compatibility equation reads:

$$-\frac{r\Phi''' + \Phi''}{\beta} + \frac{\Phi'}{r} = \frac{1}{2}(w' + w_0')^2 - \frac{1}{2}(w_0')^2. \tag{2.9}$$

By differentiating this equation once more and dividing it by $r$, we can recover an expression of Gauss's Theorema Egregium, which equates the intrinsic and extrinsic definition of Gaussian curvature, respectively. Provided that $w$ and $\Phi$ are known, the stress resultants, strains and radial displacements can be derived via equations (2.1), (2.2), (2.5) and (2.8) to calculate the total bending and stretching energy stored in the cap:

$$\left.\begin{aligned} \Pi_B &= \pi \int_0^a (\kappa_r m_r + \kappa_\theta m_\theta) r \, \mathrm{d}r \\ \text{and} \quad \Pi_S &= \pi t \int_0^a (\varepsilon_r \sigma_r + \varepsilon_\theta \sigma_\theta) r \, \mathrm{d}r + \pi a K_u u^2\big|_{r=a}, \end{aligned}\right\} \tag{2.10}$$

respectively; the latter includes a term for the contribution of a horizontal spring of stiffness $K_u$ at $r = a$.

It follows from the principle of stationary action that the strain energy of structures in equilibrium takes an extremal value, and hence, it constitutes an equipollent axiom from which equilibrium in equation (2.7) can be derived [39]. A semi-analytical energy-based approach capable of approximating alternative equilibrium configurations is presented next.

# 3. Ritz approach

We first discuss the linear solution, which is then employed in a geometrically nonlinear Ritz approach for polar-orthotropic caps (§3.2) and annuli (§3.3). We consider nonlinearity without a further linearization of the governing equations, in order to avoid a trial-and-error method when locating alternative equilibrium configurations. Hence, we limit ourselves to a few degrees of freedom but, in return, we are able to identify inverted stable configurations directly. As a consequence, our

methodology is more likely to predict 'false positives' due to either insufficient degrees of freedom or a violation of shallow shell theory (which includes the assumption of shallow gradients). In contrast to that, the FE method is prone to miss stable configurations because we cannot assess the stability of an infinite number of possible configurations.

## 3.1. Geometrically linear bending of a plate

To obtain a suitable trial function for the deflection field, $w$, let us consider the geometrically linear solution of a thin bent plate. In that case the in-plane coupling is neglected ($\Phi = 0$), and equation (2.7) simplifies to

$$w'''' + \frac{2}{r}w''' - \frac{\beta}{r^2}w'' + \frac{\beta}{r^3}w' = \frac{p}{D}. \tag{3.1}$$

The solution in the absence of load ($p = 0$) without considering rigid-body modes reads

$$w_h = \begin{cases} \text{for } \beta = 1: & A_1 r^2 + (A_2 + A_3 r^2)\log(r) \\ \text{else}: & B_1 r^{1+\sqrt{\beta}} + B_2 r^{1-\sqrt{\beta}} + B_3 r^2 \end{cases} \tag{3.2}$$

and degenerates for isotropic materials ($\beta = 1$).

The only relevant coefficient for isotropic closed shells is $A_1$ [40]. In polar-orthotropic materials, the situation is more intricate, since all constants evoke some kind of singularity when $\beta < 1$. First, we calculate the curvatures according to equation (2.1) as well as the corresponding bending moments and shear force via equations (2.5) and (2.6), respectively, and then the related bending strain energy via equation (2.10). Since $\beta > 0$, a pure deformation mode in $B_1$ has finite bending energy, whereas $B_2$ would engender an infinite energy barrier in closed shells and is thus not observed; in annular shells, however, even stress resultants containing terms in $r$ raised to powers less than −1 are energetically admissible.

The curvatures related to the $B_1$ term vary with $r$ raised to $-1 + \sqrt{\beta}$, whereas $B_3$ causes a uniform curvature (UC) throughout the shell. Note that the first term signifies a vanishing shear force throughout the shell, whereas $B_3$ causes central shear-stress singularities for $\beta \neq 1$; this rather unintuitive detail is a consequence of the material law employed. The singularity of the latter would necessitate shear-deformable Reissner–Mindlin theory, but since it is energetically favourable for thin shells to evade shear deformation by flexure, the $B_3$-term can be neglected. Hence, we assume the dominant deformation mode in the absence of load to be $B_1 r^{1+\sqrt{\beta}}$, despite causing bending-stress singularities at the centre for $\beta < 1$. The same terms arise in the solution of equation (2.9) for the Airy stress function where the $r^2$ term (technically now part of the particular solution) and $r^{1-\sqrt{\beta}}$ term vanish due to in-plane displacement compatibility and energetic admissibility in full plates, respectively.

Whether or not infinite stresses from singularities are ever acceptable in elasticity problems, we follow the philosophy of Barber [41, p. 142ff]: from a mathematical perspective, a well-posed problem has an existing, unique and converging solution. While this argument may not convince the engineering community *per se*, engineers commonly encounter and accept singularities at sharp corners and under point loads from an idealization of geometry or the boundary conditions. Consequently, knowing that results next to singularities are not applicable in practice, we choose to accept them as long as they are energetically admissible.

Here, the singularities arise directly from idealized constitutive equations. Just as there are no corners without a small fillet radius [42], perfectly polar-orthotropic materials do not exist, since fibre orientations would be undefined precisely at the singular point of $r = 0$ (cf. figure 1b–d, tantamount to a central isotropic spot). Thus, the stress definitions are predisposed for singularities. Note that singular aspects also arise when the nonlinearity of the solution is considered: according to Woinowsky-Krieger [33] singularities in stretching are encountered in Kirchhoff–Love plate theory as soon as membrane forces exist. These ultimately cause a buckled shape described by integrals of Bessel functions with concomitant singularities in bending.

## 3.2. Nonlinear solution for shallow caps

To find alternative equilibrium configurations, our choice of mode shape is inspired by the linear plate solution, but additional terms are required to satisfy the boundary conditions. Others, e.g. [43], consider an additional quartic term in consequence of a uniform pressure loading. In the present case, however,

we seek alternative *load-free* equilibria and, thus, there is no additional indicator of a preferred deformation mode that could be employed as a trial function.

An alternative approach might take its inspiration from shallow shell theory but we want to avoid intricate Bessel functions and to limit ourselves to polynomials. By considering more than one degree of freedom, we allow some latitude to mitigate the penalty of using a reasonable approximation rather than the (unknown) exact function. This increases the robustness of our methodology and allows us to cover a wider range of varying parameters. We deliberately avoid an integer power approach since some of its terms would be equivalent to terms in equation (3.2) for specific $\beta$-values. Consequently, the solution quality would deteriorate due to unsatisfied boundary conditions or a reduction of degrees of freedom.

To address these shortcomings, we assume a simple series:

$$w = A_1 + \eta_1 \rho^{1+\sqrt{\beta}} + \eta_2 \rho^{2+\sqrt{\beta}} + \eta_3 \rho^{3+\sqrt{\beta}} + A_2 \rho^{4+\sqrt{\beta}} \tag{3.3}$$

with the dimensionless radius, $\rho = r/a$ and, in total, five constants, $A_i$ and $\eta_i$. The first two, $A_1$ and $A_2$, are used to satisfy the boundary conditions of $w(\rho = 1) = 0$ and a vanishing radial bending moment at the edge, while the remaining constants, $\eta_1$, $\eta_2$, $\eta_3$ serve as degrees of freedom. The formulae for $A_i$ as well as the further particulars of the derivation of the Airy stress function are given in appendix A.

When calculating $\Phi$, the relevant homogeneous solution, $\Phi_h = C_1 a^2 \rho^{1+\sqrt{\beta}}$, indicates that stretching stresses exhibit a similar singularity as bending stresses at the centre. The constant $C_1$ is used to satisfy the boundary condition:

$$K_u u|_{\rho=1} = -\sigma_r|_{\rho=1}, \tag{3.4}$$

where $K_u$ is the stiffness of an in-plane spring, which tends in the limit to be a roller-supported boundary ($K_u = 0$) or a fixed-pinned edge ($K_u \to \infty$).

Now, the stress and strain resultants only depend on $\eta_1$, $\eta_2$ and $\eta_3$, and the energy can be calculated according to equation (2.10). Load-free equilibrium configurations are identified via

$$\nabla_\eta \Pi = 0, \tag{3.5}$$

where $\nabla_\eta$ denotes the nabla operator with respect to the three degrees of freedom. These configurations are stable, if and only if, the $3 \times 3$ stiffness matrix, **H**, with

$$H_{ij} = \frac{\partial^2 \Pi}{\partial \eta_i \partial \eta_j}, \tag{3.6}$$

is positive definite, which requires its three eigenvalues to be positive.

## 3.3. Nonlinear solution for shallow planform annuli

Even though annuli cannot have a central stress singularity, a thorough choice of the assumed deflection field is required. Following the same reasoning as before, we use the linear equilibrium solution in equation (3.2) as a part of the solution space, which now permits the second term, $B_2 \rho^{1-\sqrt{\beta}}$.

Since polynomials with negative powers are permissible now, the number of possible mode shapes increases. Taking a similar series to equation (3.3) with $\rho^{i \pm \sqrt{\beta}}$ would not allow us to satisfy the boundary condition of $u_r = 0$ for $\nu \neq 0$. Thus, we choose a slightly different expression:

$$w = \eta_{-2} \rho^{1-2\sqrt{\beta}} + \eta_{-1} \rho^{1-\sqrt{\beta}} + \eta_0 + \eta_1 \rho^{1+\sqrt{\beta}} + \eta_2 \rho^{2+\sqrt{\beta}} + \eta_3 \rho^{3+\sqrt{\beta}}. \tag{3.7}$$

Four out of the six constants are used to satisfy the boundary conditions of a hinged outer edge ($r = a$) and a free inner edge ($r = b$)

$$w|_{\rho=1} = 0, \quad m_r|_{\rho=1} = 0, \quad m_r|_{\rho=b/a} = 0 \quad \text{and} \quad q_r|_{\rho=b/a} = 0, \tag{3.8}$$

leaving the system with two degrees of freedom, $\eta_{-1}$ and $\eta_{-2}$; the expressions for $\eta_0, \eta_1, \eta_2, \eta_3$ are given in appendix A.

We do not consider more terms, since the increased number is simply not required in most cases, and considering an additional degree of freedom significantly deteriorates computational efficiency, since the deflection function is squared twice: once when computing the Airy stress function and, secondly, when calculating the stretching energy. We follow the procedure of the preceding section to compute the corresponding particular solution of the stress function, and add the homogeneous part for which a second constant of integration is now admissible, $\Phi_h = C_1 a^2 \rho^{1+\sqrt{\beta}} + C_2 a^2 \rho^{1-\sqrt{\beta}}$, in order to satisfy the

free inner edge condition, $\sigma_r = 0$ at $\rho = b/a$, and equation (3.4); details are given in appendix A. To identify alternative stable configurations, we follow the energy-minimizing procedure of §3.2.

# 4. Results

First, we present a qualitative analysis of stiffeners in §4.1 before we give a detailed analysis of the effects of polar-orthotropic materials on the inverted shape and stress resultants in §4.2. The suitability of our results is assessed by comparing them to FE calculations conducted with the commercial software *ABAQUS* [44]. In a quasi-static implicit dynamic calculation, over 900 quadratic S8R5 elements are used to model the inversion process of a quarter of a shell with circular planform with doubly-symmetric boundary conditions and properties (in SI units) of $E = 10^7$, $t = 0.01$, $a = 1$ and a density of $10^{-5}$, see [45] for details.

In §4.3, we analyse the minimum apex height required for a stable inversion as a function of the orthotropic ratio, and we present simplifying one-term approaches that capture this threshold in closed form. Verifying FE calculations were conducted with an overseeing *Python* [46] script for iterative simulations with slightly changing geometries, up to an accuracy of 0.25%. Our results inspired us to think of shells with extreme orthotropic ratios in a geometrically decoupled way, and we give a straightforward explanation using a beam analogy in §4.4. Eventually, we analyse the effects of central holes in §4.5.

## 4.1. Qualitative influence of stiffeners on bistable inversion

In general, we need to consider separate $\beta$-values for stretching, $\beta_s$, and bending, $\beta_b$, when stiffeners are added, since the stretching rigidity relates linearly to the cross-sectional height, while the flexural rigidity has a cubic relation. The small width of each stiffener imposes two free-edge conditions in close proximity that prevent stresses in the orthogonal direction (figure 2). Thus, the approximate orthotropic parameters stem from the area of stiffeners in their longitudinal direction.

For deformed shells of finite thickness, relaxation of the internal bending stresses always tries to restore the shell back to its initial configuration but which may be prevented by a stretching barrier. Additional stiffeners therefore tend to erode bistability by increasing the bending rigidity disproportionally, while carving out or removing material to weaken a particular direction tends to favour bistability. This aspect reduces to a discussion that is more about an effective thickness than the problem of directional stiffness, and is not further elaborated since it is well known that the bistable threshold of the initial apex height scales with the thickness [47]. If, however, a detailed quantitative analysis is desired, one can calculate the linear bending solution, on which the assumption of the deflection field is based, using $\beta_b$, while the homogeneous terms of the Airy stress function depend on $\beta_s$; ways to approximate the particular values of $\beta_b$ and $\beta_s$ are, for instance, described by Ventsel & Krauthammer [48].

## 4.2. Quantitative analysis: inverted shapes and corresponding stress resultants

Setting $\beta = \beta_s = \beta_b$, we depict stable inverted configurations for pinned and roller-supported shells in figure 3$a$,$b$, respectively, for the indicated values of $\beta$. All shells have the same initial height of $w_0^M/t = 4$, which ensures that all cases in (a) exhibit bistability, where roller-supported shells of that height are bistable for the range $0.5 < \beta < 6.1$.

For $\beta < 1$, displacements are more focused at the centre, and increasing $\beta$ shifts the deformation towards the outer regions, so that shells with $\beta \gtrsim 3$ evince an inflexion point *viz.* a central dimple. Both responses confirm the observed behaviour in the stiffened shells in figure 1$d$ and $b/c$, respectively. Note that smaller $\beta$-values do not always correspond with larger central deformations since the roller-supported case with $\beta = 0.5$ has a decreased, yet centrally more focused, deformation than the corresponding isotropic case.

While a concentrated deformation points towards highly stressed areas, the barely deformed central region of the dimple indicates low bending stresses. Correspondingly, the resulting stresses, depicted by solid lines in figure 4 for $\beta = 0.1, 0.5, 1, 5, 10$, are absent at the very centre for $\beta > 1$. These are in good agreement with FE results (dots), whereas even *higher-order* approaches of integer power (based on an approach for isotropic shells in [34], dashed line) shows slight deviations; note that lower-order integer approaches that apply simpler basis functions to polar-orthotropic shells, e.g. [49–51], lead to less accurate results. A transition point with finite central stresses is encountered for the degenerate case of $\beta = 1$, where the integer power approach coincides with equation (3.3). Below this value ($\beta < 1$),

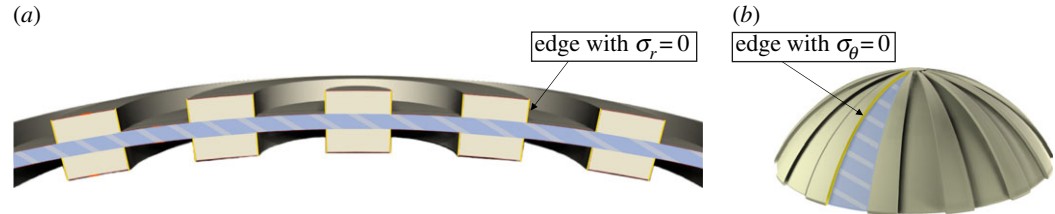

**Figure 2.** (a) Sectional view of a circumferentially stiffened shell: approximated effective areas in radial (blue hatched) and circumferential direction (beige + blue hatched). The stiffened area is neglected in radial direction, since stresses cannot evenly distribute through the stiffener's width due to the free edge boundary conditions of $\sigma_r = 0$ (yellow lines). (b) Full view of a radially stiffened shell: highlights exemplify now the effective area in circumferential direction (blue hatched) and the free edge condition $\sigma_\theta = 0$ (yellow); here, the full cross section of ribs is only considered during the calculation of the smeared stiffness in radial direction.

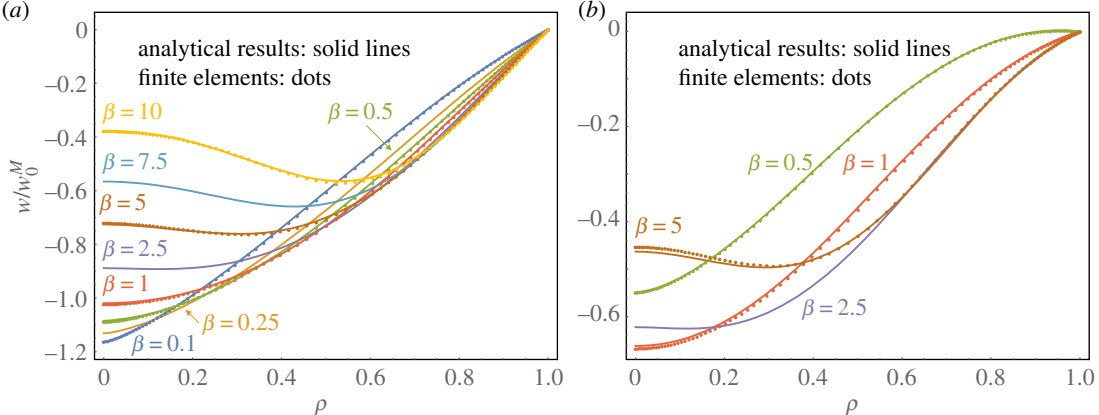

**Figure 3.** Sectional view of stable inverted shapes normalized by $w_0^M = 4t$ for (a) fixed-pinned and (b) roller-supported edges with $0.1 \le \beta \le 10$ for $\nu = 0$; analytical predications (solid lines) and FE results (dots). The right picture relates to the observations in figure 1 and confirms that centralized deformations occur for $\beta < 1$, whereas shells with $\beta \gtrsim 3$ evince a central dimple.

FE calculations confirm induced stress singularities in bending and stretching, which are accurately captured by our analytical model using equation (3.3).

It now becomes apparent that integer power approaches are inferior since they only capture singularities in stretching but not in bending, which underestimates peak stresses. The loss of accuracy cannot be overcome by increasing the number of degrees of freedom since the polynomial order does not match. A closer inspection of the central region in a doubly logarithmic plot of $m_\theta$ in figure 5 shows that the use of real powers in equation (3.3) accurately captures the asymptotic behaviour for $\rho \ll 1$, where the approximately linear relation in the diagram confirms the dominating influence of the $\rho^{-1+\sqrt{\beta}}$ term. This inspires us to employ the related deflection term in a simplified single degree-of-freedom approach to predict the bistable threshold, which is discussed next.

## 4.3. Minimum apex height required for bistable inversion: refined approaches and simplifications

Geometrical thresholds for bistable inversion are indicated by a prime; the non-dimensional initial apex height, $\omega_{0*}^M = w_{0*}^M / t$, is given as a function of $\beta$ for various choices of the deflection field in figure 6, for roller-supported and fixed-pinned edges.

In general, in-plane supports strongly favour bistable inversion, which confirms a recent observation in [34]. More interestingly, the influence of the stiffness ratio differs significantly, depending on the support conditions. For a pinned edge, smaller values of $\beta$ seem to generally favour bistable inversion, while the same values for roller-supported shells hamper and eventually erode bistable behaviour altogether. A global minimum in the latter case is found for $\beta = 3.2$, which coincides approximately with the $\beta$-ratio at which the deflection field is about to first form an inflexion point in figure 3.

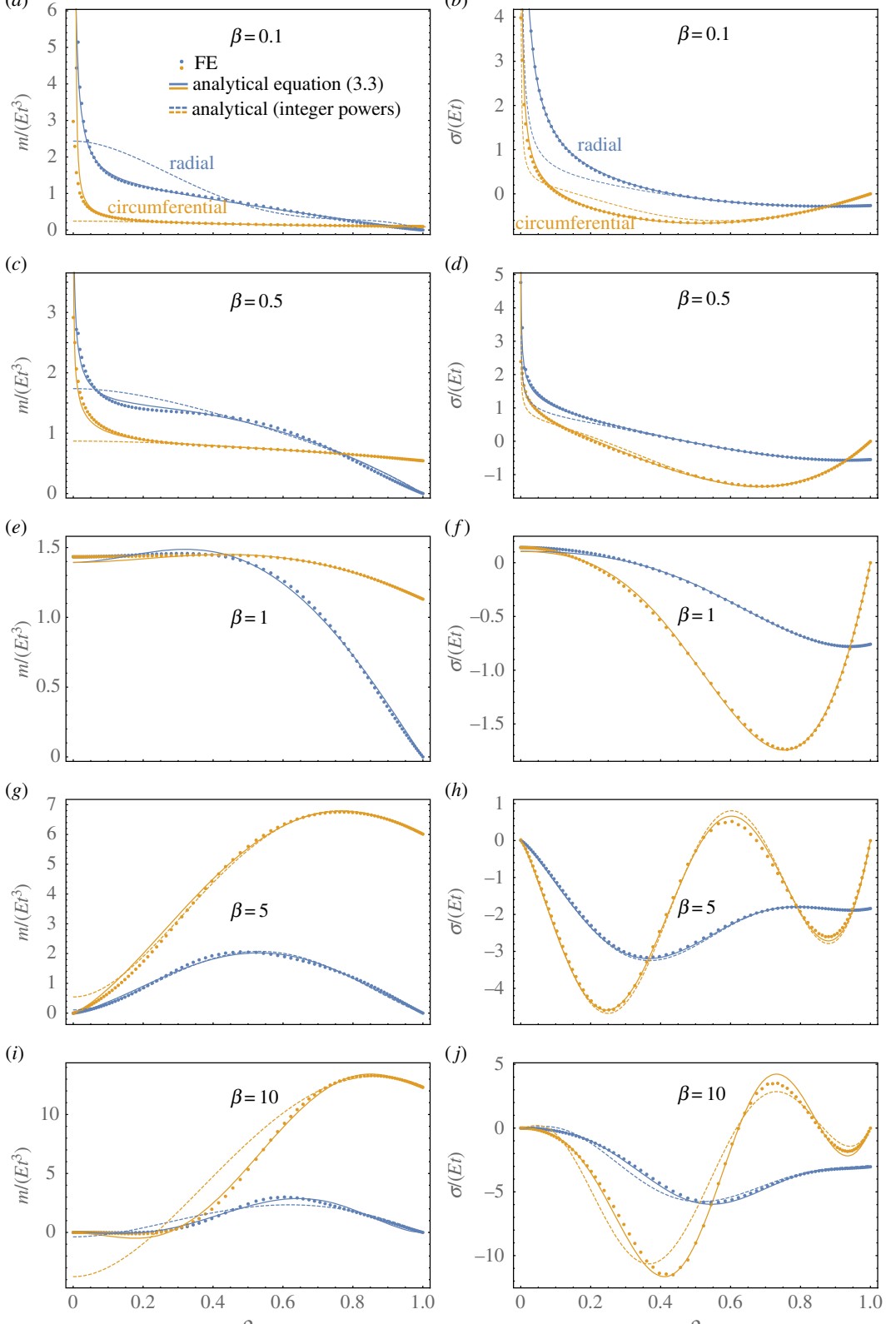

**Figure 4.** Bending $(a,c,e,g,i)$ and stretching $(b,d,f,h,j)$ stress resultants for differing stiffness-ratios of a fixed-pinned shell with $w_0^M = 4t$ and $\nu = 0$.

In terms of computational accuracy, the results are virtually indistinguishable from FE results, with an average deviation of 0.35%, whereas the FE accuracy range was set to 0.25%. The approximation is superior to results obtained by adapted lower-order models from the literature with a single degree of

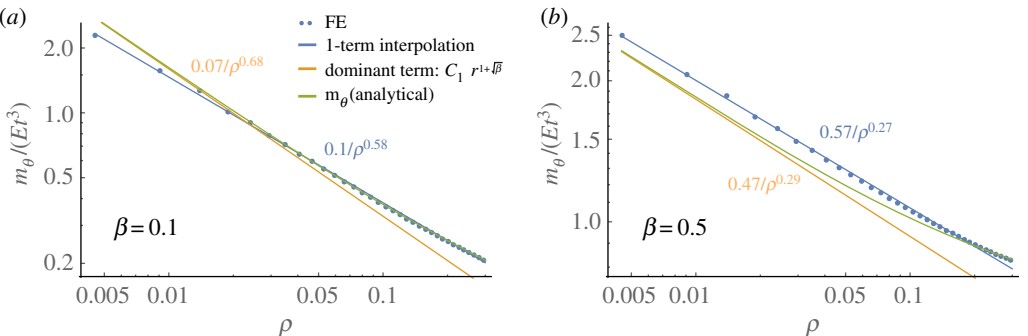

**Figure 5.** Influence of linear bending term, $\rho^{-1+\sqrt{\beta}}$ for $\beta < 1$: FE results (dots) of circumferential bending moments in inverted configuration on doubly logarithmic scale with linear-reciprocal regression (blue line) are compared to prediction of the unsimplified model (green) and the dominant term that is related to the linear bending solution in equation (3.2), $C_1 \rho^{-1+\sqrt{\beta}}$. Examples are depicted for $\beta = 0.1$ (a) and $\beta = 0.5$ (b) with $w_0^M = 4t$, $\nu = 0$ and fixed-pinned edges.

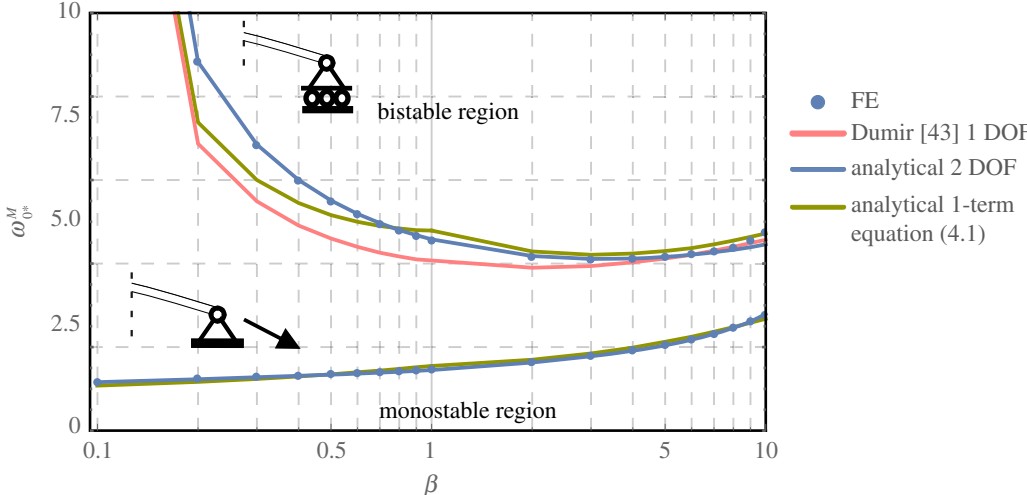

**Figure 6.** Predictions of the critical dimensionless initial apex height, $\omega_{0*}^M$, over stiffness ratio, $\beta$, for roller-supported (top) and fixed-pinned supports (bottom) for $\nu = 0.3$. Solid lines indicate analytical predictions, while dots represent results from FE simulations. For pinned supports only the current approach in equation (3.3) and the simplified one-term approach according to equation (4.1) are compared to FE results for the sake of clarity.

freedom, e.g. by Dumir [43], which shows an average deviation of 5.2%. Closed-form solutions are found by simplifying one-term assumptions of $w = \eta_1 \rho^{1+\sqrt{\beta}}$ for $\beta < 1$ and $w = \eta_1 \rho^2$ for $\beta \geq 1$:

$$
\left.
\begin{aligned}
(\omega_{0*}^M)^2 &= \frac{(1+\sqrt{\beta})^4(2+\sqrt{\beta})(2-\nu+5\sqrt{\beta})}{3[16\beta^{3/2}+2\beta^2+\beta(20-8\nu)-(3-\nu)^2\sqrt{\beta}+2]} && \text{for } \beta < 1 \\
(\omega_{0*}^M)^2 &= \frac{(3+\sqrt{\beta})^2(1+2\nu+\beta)}{2[\beta+6\sqrt{\beta}-\nu(\nu-6)]} && \text{for } \beta \geq 1
\end{aligned}
\right\} \text{pinned}
$$

$$
\left.
\begin{aligned}
(\omega_{0*}^M)^2 &= \frac{(1+\sqrt{\beta})^4(2+\sqrt{\beta})}{3(\beta-\sqrt{\beta}\nu)} && \text{for } \beta < 1 \\
(\omega_{0*}^M)^2 &= \frac{(3+\sqrt{\beta})^2(1+2\nu+\beta)}{2(\beta-\nu^2)} && \text{for } \beta \geq 1
\end{aligned}
\right\} \text{rollers.}
\tag{4.1}
$$

These results emphasize the importance of the transition around $\beta = 1$: for small values, the deflection is governed by the homogeneous term of the linear bending solution while, surprisingly, the averaging nature of a UC approach can predict the stability threshold for $\beta \geq 1$ despite the clearly non-uniform displacement field in figure 3.

## 4.4. The beam analogy

The dependency on the boundary conditions considers the limits of $\beta \to \nu^2$ and $\beta \to \infty$. Furthermore, in the first case, if we assume $\nu = 0$ (and $\beta \to 0$) for simplicity, the circumferential stiffness tends towards zero

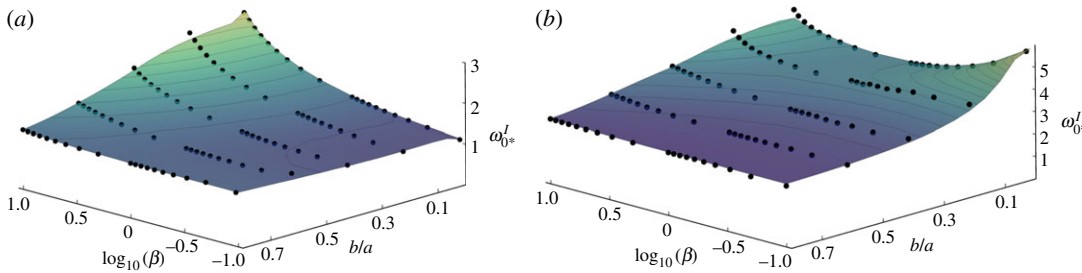

**Figure 7.** Stability map of critical initial dimensionless physical height, $\omega^I_{0*}$, over central hole size, $b/a$, and stiffness ratio with $\nu = 0$ for (a) pinned and (b) roller-supported edges; dots indicate FE results.

and the first two terms of the compatibility equation, equation (2.9), tend to infinity. Hence, large changes in Gaussian curvature do not induce any stresses, which is reasonable since the shell is virtually free to expand or contract in the circumferential direction. Consequently, the shell response resembles a symmetric beam with a wedge-planform of vanishing width at the centre, similar to the shape of a single stiffener in figure 2b.

The compatibility equation then becomes a simplified version of equation (2.9):

$$\sigma_r = \frac{\phi'}{r} = \frac{1}{2}(w' + w'_0)^2 - \frac{1}{2}(w'_0)^2, \tag{4.2}$$

and reflects the entirely geometric strain relation in equation (2.8) without hoop-interaction. Singularities, however, do arise because the area of the tapered cross section vanishes at the centre, even though the radial *force* is well defined. Interestingly, the bistable threshold of fixed-pinned shells with $\beta = 0.1$ precisely matches the threshold of fixed-pinned beams, of $\omega^M_{0*} = 1.1$.

In this decoupled case, the only way to cause a stretching barrier that inhibits the structure's bending recovery to the initial configuration is by additional horizontal supports, otherwise the structure is statically determinate. We observe a similar behaviour in shells with $\beta \ll 1$, where the structure avoids significant radial stresses by circumferential expansion. Note that up to a value of $\beta \approx 3.2$, an increasing circumferential stiffness also exerts a contrasting influence depending on the boundary conditions: while vital in case of roller supports, it becomes redundant and even hinders once a stabilizing radial force is assured by an immovable support.

In the case of $\beta \to \infty$, the radial stiffness becomes negligibly small but unlike before, the equations are not entirely decoupled, since we may think of our shell as multiple adjacent ring beams whose radial displacements interact with circumferential strains via $\varepsilon_\theta = u/r$. This interaction ensures a certain degree of statical indeterminacy and, hence, we observe a bistable response in roller-supported shells even for large values of $\beta$.

## 4.5. Bistable annular inversion

By cutting a hole, the value of the critical initial midpoint displacement, $\omega^M_{0*}$, becomes less representative as it describes an *imaginary* height of a shell before the hole was created. For our uniformly curved, closed shells, it relates to a measure of curvature via $2\omega^M_{0*} = \kappa_{0*} \cdot a^2/t$, which is a commonly used quantity to express the bistable threshold [16,17,22]. However, in the novel context of annular shells, the minimum *physical* height of the *inner* edge required for bistable inversion, $\omega^I_{0*} = (1 - b^2/a^2)\omega^M_{0*}$, is more relevant for practical applications.

The influence of a central hole on this parameter as a function of $\beta$ is presented in figure 7 for (a) fixed-pinned and (b) roller-supported edges. For a more open perspective on both plots, we reversed the axes and plotted the values of $\log_{10}\beta$ from −1 to 1, in order to cover the same range as before ($0.1 \leq \beta \leq 10$). Note that the smallest hole size calculated from the annular model was $b/a = 0.05$; the results of the closed-shell model have been added for $b/a = 0$ and intermediate values were linearly interpolated; hence, there are slight kinks in the transition zone because of the slightly different choice of basis functions, see equations (3.3) and (3.7).

Our focus is on the interdependency of the stiffness ratio $\beta$ and the hole size, $b/a$ with respect to their effect on the critical height for bistable inversion, $\omega^I_{0*}$. Poisson's ratio was set to zero so that the smallest value of $\beta = 0.1$ in figure 7 has a less hindering influence on bistability.

It can be seen that creating a hole favours bistable inversion in all shells on rollers as well as in most shells with fixed-pinned edges. The most significant enhancement of bistability is observed in roller-supported shells with a small hoop stiffness, figure 7b, where a central hole of $b/a = 20\%$ reduces $\omega^I_{0*}$

by over 30% for shells with $\beta = 0.1$; by contrast, an isotropic shell ($\beta = 1$) improves only by 3% when a hole of the same size is created.

In fixed-pinned shells, figure 7a, a similar reduction is observed with the exception of a slight increase of the physical height required for small values of $\beta$. Note that the smallest required height is actually found in a full plate with $\beta \rightarrow 0$. For large hole-sizes shell ($b/a > 0.5$), all shells begin to resemble a (doubly) curved beam with virtually no radial stresses. Since these structures are already 'decoupled' geometrically, the radial stress components are of diminished relevance.

# 5. Summary and conclusion

We have presented a higher-order, geometrically nonlinear analytical model with up to three degrees of freedom, in order to study bistable behaviour of polar-orthotropic shells. A Ritz approach that relates the deflection field to in-plane stresses via Gauss's Theorema Egregium has been used to find alternative stable configurations; these were identified by solving a nonlinear eigenvalue problem whose solution unveiled minima in the strain energy functional. Our analytical model has appropriately captured central stress singularities for low circumferential stiffness, $\beta < 1$, demonstrating its superiority in comparison to integer-power approaches.

The critical apex height required for bistable inversion has been studied for different support conditions. Using simplified averaging single-term approaches has allowed us to capture the bistable threshold in closed form, even though they fail to predict stress resultants appropriately. It has been found that a fixed-pinned edge enhances bistability in all observed examples compared to a roller-supported edge. Furthermore, the support conditions significantly affect the influence of other parameters, such as $\beta$ and the hole size: in contrast to roller-supported shells, where no alternative equilibrium configurations were found for $\beta \rightarrow 0$, fixed-pinned shells have the lowest required apex height. This difference is caused by the increased statical indeterminacy of shells, which was identified as key for stable inversion.

For roller supports, we have confirmed observations in the literature [52] that circumferential stresses have a stabilizing influence and prevent reversion. However, the opposite is true for the fixed-pinned case, where an increasing $\beta$-ratio hampers bistability. We conclude that the circumferential stiffness is stabilizing insofar as radial stresses arise from a ring beam effect that ensures a higher degree of statical indeterminacy. If radial stresses are assured by the support conditions, the circumferential stiffness is a redundant feature that becomes a slight impediment.

To circumvent stress singularities for $\beta < 1$, central holes were considered. While large holes increase the critical physical height of fixed-pinned shells only slightly, roller-supported ones were significantly more inclined to stay inverted once a circular hole of around 20% of the outer radius was formed.

Data accessibility. This work does not have any physical experimental data. The essential codes for FE analysis [45] via ABAQUS [44] and for solving the analytical models [53] via Mathematica [54] are available in the cited repositories and for open consultation, which also contain the data for generating the results presented in figures 3–7.

Authors' contributions. P.M.S. and K.A.S. conceived the methodology, interpreted the results, and wrote the paper in equal measure. P.M.S. performed the FE simulations and wrote the Mathematica code, all in consultation with K.A.S. Both authors gave final approval for publication.

Competing interests. We declare we have no competing interests.

Funding. P.M.S. is grateful to the Friedrich-Ebert-Foundation for their financial support; this work was otherwise not supported by funding from any other agency.

Acknowledgements. Both authors are grateful to their colleagues Dr Fumiya Iida and Josephine Hughes for their generous help in three-dimensional printing the moulds in which the stiffened shells were cast. The authors are also grateful to the four anonymous reviewers for their constructive feedback.

# Appendix A. Full cap

To satisfy the boundary conditions of $w(a) = 0$, $m_r(a) = 0$, substitute the following values in equation (3.3):

$$\left.\begin{array}{c} A_1 = -a^{1+\sqrt{\beta}}(\eta_1 + a\eta_2 + a^2\eta_3 + a^3\eta_4) \\[2mm] \text{and} \quad A_4 = \dfrac{\eta_1 a(1 + \sqrt{\beta})(\sqrt{\beta} + \nu) + \eta_2 a^2(2 + \sqrt{\beta})(1 + \sqrt{\beta} + \nu) + \eta_3 a^3(3 + \sqrt{\beta})(2 + \sqrt{\beta} + \nu)}{-a^4(4 + \sqrt{\beta})(3 + \sqrt{\beta} + \nu)} . \end{array}\right\} \quad (A\,1)$$

To account for the degenerate case of $\beta = 1$, we have to consider the term $\eta_2\rho^3$ causing a non-vanishing shear-force at the centre. For this particular case, we substitute this term with the next one in the series in equation (3.3), $\eta_2\rho^6$, and achieve results barely distinguishable from those in our isotropic study [34].

Using the compatibility equation (2.9), the Airy stress function, $\Phi = \Phi_p + \Phi_h$, is expressed in terms of the $\eta$ constants as

$$
\begin{aligned}
\Phi_p = \frac{E\beta}{2}\sum_{i=1}^{4}\sum_{j=1}^{4}\left(\sqrt{\beta}+i\right)\eta_i\rho^{\sqrt{\beta}+i}\Bigg[&\frac{4w_0^M\rho^2\delta_{ij}}{(i+1)(\sqrt{\beta}+i+2)(2\sqrt{\beta}+i+1)}\\
+&\frac{(\sqrt{\beta}+j)\eta_j\rho^{\sqrt{\beta}+j}}{(\sqrt{\beta}+i+j-1)(2\sqrt{\beta}+i+j)(3\sqrt{\beta}+i+j-1)}\Bigg]
\end{aligned}
\tag{A 2}
$$

where $\delta_{ij}$ denotes the Kronecker delta. Denoting the radial and circumferential stresses arising from the particular solution, $\Phi_p$, with $\sigma_{pr}$ and $\sigma_{p\theta}$, the remaining constant takes the value

$$
C_1 = -\frac{\sigma_{pr}(\nu K_u a + \beta E) - K_u a\sigma_{p\theta}}{(\sqrt{\beta}+1)[K_u a(\nu - \sqrt{\beta}) + \beta E]}\Bigg|_{\rho=1},
\tag{A 3}
$$

which simplifies to

$$
C_1 = \frac{-\sigma_{pr}}{1+\sqrt{\beta}}\Bigg|_{\rho=1} \quad \begin{array}{c}\text{for } K_u = 0\\ \text{(rollers)}\end{array} \quad \text{or} \quad C_1 = -\frac{\sigma_{p\theta}-\nu\sigma_{pr}}{(1+\sqrt{\beta})(\sqrt{\beta}-\nu)}\Bigg|_{\rho=1} \quad \begin{array}{c}\text{for } K_u \to \infty\\ \text{(fixed pins)}\end{array}.
\tag{A 4}
$$

# Appendix B. Annulus

To satisfy the boundary conditions of $w(a) = 0$, $m_r(a) = 0$, $m_r(b) = 0$ and $q_r(b) = 0$ substitute the following values in equation (3.7) one after another:

$$
\begin{aligned}
\eta_0 &= -(\eta_{-2} + \eta_{-1} + \eta_1 + \eta_2 + \eta_3),\\
\eta_3 &= \frac{b^{-5\sqrt{\beta}}(3\eta_{-2}(2\sqrt{\beta}-1)a^{5\sqrt{\beta}} - 3\eta_2(2\sqrt{\beta}+1)a^{\sqrt{\beta}}b^{4\sqrt{\beta}})}{8(3\sqrt{\beta}+1)},\\
\eta_2 &= \big[-8b^{5\sqrt{\beta}}(\eta_{-2}(4\beta+\nu-2(\nu+1)\sqrt{\beta}) + \eta_{-1}(\beta+\nu-(\nu+1)\sqrt{\beta}) + \eta_1(\beta+\nu+\nu\sqrt{\beta}+\sqrt{\beta}))\\
&\quad -3\eta_{-2}(2\sqrt{\beta}-1)a^{5\sqrt{\beta}}(\nu+3\sqrt{\beta})\big]\big/\big[b^{4\sqrt{\beta}}((2\sqrt{\beta}+1)(8b^{\sqrt{\beta}}(\nu+2\sqrt{\beta}) - 3a^{\sqrt{\beta}}(\nu+3\sqrt{\beta})))\big]\\
\text{and}\quad \eta_1 &= \big[a^{3\sqrt{\beta}}b^{\sqrt{\beta}}(\eta_{-2}(-28\beta^{3/2} + \beta(14-6\nu) + \nu(-5\nu+10\nu\sqrt{\beta}+3\sqrt{\beta}))\\
&\quad + 3\eta_{-1}(3\beta^{3/2} - \beta(2\nu+3) + \nu(\nu - \nu\sqrt{\beta}+2\sqrt{\beta}))) - 12\eta_{-2}a^{4\sqrt{\beta}}(\sqrt{\beta}-2\beta)(\nu+3\sqrt{\beta})\\
&\quad + a^{\sqrt{\beta}}(-b^{3\sqrt{\beta}})(5\nu+7\sqrt{\beta})(\eta_{-2}(4\beta+\nu-2(\nu+1)\sqrt{\beta}) + \eta_{-1}(\beta+\nu-(\nu+1)\sqrt{\beta}))\\
&\quad -(5\nu+7\sqrt{\beta})((ab)^{2\sqrt{\beta}}+b^{4\sqrt{\beta}})(\eta_{-2}(4\beta+\nu-2(\nu+1)\sqrt{\beta}) + \eta_{-1}(\beta+\nu\\
&\quad -(\nu+1)\sqrt{\beta}))\big]\big/\big[b^{3\sqrt{\beta}}((\sqrt{\beta}+1)(\nu+\sqrt{\beta})(b^{\sqrt{\beta}}(5\nu+7\sqrt{\beta}) - 3a^{\sqrt{\beta}}(\nu+3\sqrt{\beta})))\big].
\end{aligned}
\tag{A 5}
$$

Then use the same substitution to calculate the Airy stress function in terms of the remaining two degrees of freedom, $\eta_{-2}$ and $\eta_{-1}$:

$$
\begin{aligned}
\Phi'_p = -\frac{E\beta}{2a}\sum_{i=-2}^{3}\sum_{j=-2}^{3}(1+i\sqrt{\beta})\rho^{1+i\sqrt{\beta}}\eta_i\Bigg[&\frac{4w_0^M\rho\,\delta_{ij}}{(2+\sqrt{\beta}+i\sqrt{\beta})(2+(i-1)\sqrt{\beta})}\\
+&\frac{(1+j\sqrt{\beta})\rho^{j\sqrt{\beta}}\eta_j(1-\delta_{-ij})}{(1+(i+j-1)\sqrt{\beta})(1+(1+i+j)\sqrt{\beta})}\Bigg](1-\delta_{i0})(1-\delta_{j0})
\end{aligned}
\tag{A 6}
$$

The constants for annulus with $K_u \to 0$ (roller supports) are

$$
C_1 = \frac{\sigma_{pr|\rho=1} - (b/a)^{1+\sqrt{\beta}}\sigma_{pr|\rho=b/a}}{(1+\sqrt{\beta})[(b/a)^{2\sqrt{\beta}}-1]}
\tag{A 7}
$$

and

$$
C_2 = -\frac{(b/a)^{\sqrt{\beta}}[(b/a)^{\sqrt{\beta}}\sigma_{pr|\rho=1} - (b/a)\sigma_{pr|\rho=b/a}]}{(1+\sqrt{\beta})[(b/a)^{2\sqrt{\beta}}-1]}
$$

and the constants for annulus with $K_u \to \infty$ (fixed pins) are

$$
\left.\begin{aligned}
C_1 &= \frac{a^{1+\sqrt{\beta}}(\nu\sigma_{pr}(a) - \sigma_{p\theta}(a)) - b^{1+\sqrt{\beta}}(\sqrt{\beta} + \nu)\sigma_{pr}(b)}{(\sqrt{\beta} + 1)(a^{2\sqrt{\beta}}(\sqrt{\beta} - \nu) + b^{2\sqrt{\beta}}(\sqrt{\beta} + \nu))} \\
\text{and} \quad C_2 &= \frac{a^{\sqrt{\beta}}b^{\sqrt{\beta}}(ba^{\sqrt{\beta}}(\sqrt{\beta} - \nu)\sigma_{pr}(b) + ab^{\sqrt{\beta}}[\nu\sigma_{pr}(a) - \sigma_{p\theta}(a)])}{(\sqrt{\beta} - 1)[a^{2\sqrt{\beta}}(\sqrt{\beta} - \nu) + b^{2\sqrt{\beta}}(\sqrt{\beta} + \nu)]}
\end{aligned}\right\}
\tag{A 8}
$$

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
