## [Reviewer comments · Royal Society Open Science]

Review History

RSOS-190888.R0 (Original submission)

Review form: Reviewer 1

Is the manuscript scientifically sound in its present form?

Yes

Are the interpretations and conclusions justified by the results?

Yes

Is the language acceptable?

Yes

Is it clear how to access all supporting data?

Not Applicable

Do you have any ethical concerns with this paper?

No

Have you any concerns about statistical analyses in this paper?

No

Recommendation?

Accept as is

Comments to the Author(s)

The authors have taken my comments into consideration and in my opinion the manuscript has been improved significantly. I fully support publication.

Minor remarks:

*Poisson's is misspelled as Poission's below Eq 2

*Replace 'extreme' with 'extremal' or 'extremum' below Eq 10

*There is an unformatted latex command /times left above eq 16

Review form: Reviewer 2

Is the manuscript scientifically sound in its present form?

Yes

Are the interpretations and conclusions justified by the results?

Yes

Is the language acceptable?

Yes

Is it clear how to access all supporting data?

Yes

Do you have any ethical concerns with this paper?

No

Have you any concerns about statistical analyses in this paper?

No

Recommendation?

Accept with minor revision (please list in comments)

Comments to the Author(s)

The authors have addressed my comments from the previously submitted version satisfactorily.

However, I have two minor further comments on their responses that they may wish to consider:

1. In the final sentence of the abstract, is it clear what "it" refers to? Perhaps they should say "whilst fixed-pinned shells are barely affected by a hole, the presence of a hole strongly favours bistable inversions in roller-supported shells".

2. I find the labelling "Remark 1" a bit confusing, since it still suggests "Remark 2" is coming - the later discussion of singularities is not labelled in the same way. Perhaps they could just not number the remark?

Decision letter (RSOS-190888.R0)

24-Jun-2019

Dear Mr Sobota

On behalf of the Editors, I am pleased to inform you that your Manuscript RSOS-190888 entitled "Bistable polar-orthotropic shallow shells" has been accepted for publication in Royal Society Open Science subject to minor revision in accordance with the referee suggestions. Please find the referees' comments at the end of this email.

The reviewers and handling editors have recommended publication, but also suggest some minor revisions to your manuscript. Therefore, I invite you to respond to the comments and revise your manuscript.

- Ethics statement

- Data accessibility

<http://datadryad.org/submit?journalID=RSOS&manu=RSOS-190888>

- Competing interests

- Authors' contributions

AB carried out the molecular lab work, participated in data analysis, carried out sequence alignments, participated in the design of the study and drafted the manuscript; CD carried out

the statistical analyses; EF collected field data; GH conceived of the study, designed the study, coordinated the study and helped draft the manuscript. All authors gave final approval for publication.

- Acknowledgements

- Funding statement

Because the schedule for publication is very tight, it is a condition of publication that you submit the revised version of your manuscript before 03-Jul-2019. Please note that the revision deadline will expire at 00.00am on this date. If you do not think you will be able to meet this date please let me know immediately.

- 1) A text file of the manuscript (tex, txt, rtf, docx or doc), references, tables (including captions) and figure captions. Do not upload a PDF as your "Main Document";
- 2) A separate electronic file of each figure (EPS or print-quality PDF preferred (either format should be produced directly from original creation package), or original software format);
- 3) Included a 100 word media summary of your paper when requested at submission. Please ensure you have entered correct contact details (email, institution and telephone) in your user account;
- 4) Included the raw data to support the claims made in your paper. You can either include your data as electronic supplementary material or upload to a repository and include the relevant doi within your manuscript. Make sure it is clear in your data accessibility statement how the data can be accessed;

5) All supplementary materials accompanying an accepted article will be treated as in their final form. Note that the Royal Society will neither edit nor typeset supplementary material and it will be hosted as provided. Please ensure that the supplementary material includes the paper details where possible (authors, article title, journal name).

on behalf of R. Kerry Rowe (Subject Editor)
openscience@royalsociety.org

Associate Editor Comments to Author:

Thanks for submitting this transfer to Royal Society Open Science. Two of the reviewers from your earlier Proceedings A submission have kindly reviewed your revised manuscript. As you will note they are broadly in favour of publication but offer a number of minor modifications to improve readability of the manuscript. We would recommend you incorporate these changes and provide a point-by-point response indicating that you have done so. Congratulations on this otherwise fine work.

Reviewer comments to Author:
Reviewer: 1

Comments to the Author(s)

The authors have taken my comments into consideration and in my opinion the manuscript has been improved significantly. I fully support publication.

Minor remarks:

- *Poisson's is misspelled as Poission's below Eq 2
- *Replace 'extreme' with 'extremal' or 'extremum' below Eq 10
- *There is an unformatted latex command /times left above eq 16

Reviewer: 2

Comments to the Author(s)

The authors have addressed my comments from the previously submitted version satisfactorily.

However, I have two minor further comments on their responses that they may wish to consider:

1. In the final sentence of the abstract, is it clear what "it" refers to? Perhaps they should say "whilst fixed-pinned shells are barely affected by a hole, the presence of a hole strongly favours bistable inversions in roller-supported shells".
2. I find the labelling "Remark 1" a bit confusing, since it still suggests "Remark 2" is coming - the later discussion of singularities is not labelled in the same way. Perhaps they could just not number the remark?

Author's Response to Decision Letter for (RSOS-190888.R0)

See Appendix A.

Decision letter (RSOS-190888.R1)

04-Jul-2019

Dear Mr Sobota,

I am pleased to inform you that your manuscript entitled "Bistable polar-orthotropic shallow shells" is now accepted for publication in Royal Society Open Science.

Kind regards,
Andrew Dunn
Senior Publishing Editor
Royal Society Open Science
openscience@royalsociety.org

on behalf of Prof R. Kerry Rowe (Subject Editor)
openscience@royalsociety.org

Follow Royal Society Publishing on Twitter: [@RSocPublishing](https://twitter.com/RSocPublishing)

Appendix A

Dear editor,
please find enclosed the revision of our manuscript "Bistable Polar-Orthotropic Shallow Shells".

All suggestions of the reviewers were implemented in the manuscript and highlighted in red colour; previous changes are highlighted in blue, as before.

Yours sincerely,

Paul Sobota & Keith Seffen

Summary of changes:

1. The final sentence of the abstract now reads: "whilst fixed-pinned shells are barely affected by a hole, *the presence of a hole* strongly favours bistable inversions in roller-supported shells".
2. Changed "Remark 1" to "Remark" (without a number)
3. Corrected a typo below Eq 2 ("Poisson's ratio")
4. Replaced 'extreme' with 'extremal' below Eq 10, as suggested.
5. Corrected an unformatted latex command above Eq 16